# Algae Food Products as a Healthcare Solution

**DOI:** 10.3390/md21110578

**Published:** 2023-11-05

**Authors:** Joana O. Tavares, João Cotas, Ana Valado, Leonel Pereira

**Affiliations:** 1Department of Life Sciences, University of Coimbra, 3000-456 Coimbra, Portugal; joana02tavares@gmail.com (J.O.T.); jcotas@uc.pt (J.C.); 2MARE—Marine and Environmental Sciences Centre/ARNET-Aquatic Research Network, University of Coimbra, 3000-456 Coimbra, Portugal; valado@estesc.ipc.pt; 3Biomedical Laboratory Sciences, Coimbra Health School, Polytechnic Institute of Coimbra, Rua 5 de Outubro—SM Bispo, Apartado 7006, 3045-043 Coimbra, Portugal; 4Instituto do Ambiente Tecnologia e Vida, Faculdade de Ciências e Tecnologia, Rua Sílvio Lima, 3030-790 Coimbra, Portugal

**Keywords:** seaweed, bioactive, nutraceuticals, celiac disease, diabetes, hyperglycemia, glycemic index, cholesterol management

## Abstract

Diseases such as obesity; cardiovascular diseases such as high blood pressure, myocardial infarction and stroke; digestive diseases such as celiac disease; certain types of cancer and osteoporosis are related to food. On the other hand, as the world’s population increases, the ability of the current food production system to produce food consistently is at risk. As a result, intensive agriculture has contributed to climate change and a major environmental impact. Research is, therefore, needed to find new sustainable food sources. One of the most promising sources of sustainable food raw materials is macroalgae. Algae are crucial to solving this nutritional deficiency because they are abundant in bioactive substances that have been shown to combat diseases such as hyperglycemia, diabetes, obesity, metabolic disorders, neurodegenerative diseases and cardiovascular diseases. Examples of these substances include polysaccharides such as alginate, fucoidan, agar and carrageenan; proteins such as phycobiliproteins; carotenoids such as β-carotene and fucoxanthin; phenolic compounds; vitamins and minerals. Seaweed is already considered a nutraceutical food since it has higher protein values than legumes and soy and is, therefore, becoming increasingly common. On the other hand, compounds such as polysaccharides extracted from seaweed are already used in the food industry as thickening agents and stabilizers to improve the quality of the final product and to extend its shelf life; they have also demonstrated antidiabetic effects. Among the other bioactive compounds present in macroalgae, phenolic compounds, pigments, carotenoids and fatty acids stand out due to their different bioactive properties, such as antidiabetics, antimicrobials and antioxidants, which are important in the treatment or control of diseases such as diabetes, cholesterol, hyperglycemia and cardiovascular diseases. That said, there have already been some studies in which macroalgae (red, green and brown) have been incorporated into certain foods, but studies on gluten-free products are still scarce, as only the potential use of macroalgae for this type of product is considered. Considering the aforementioned issues, this review aims to analyze how macroalgae can be incorporated into foods or used as a food supplement, as well as to describe the bioactive compounds they contain, which have beneficial properties for human health. In this way, the potential of macroalgae-based products in eminent diseases, such as celiac disease, or in more common diseases, such as diabetes and cholesterol complications, can be seen.

## 1. Introduction

Overweight/obesity, cardiovascular illnesses (such as arterial hypertension, myocardial infarction and stroke), diabetes mellitus, digestive diseases (such as celiac disease), certain malignancies and osteoporosis are all diet-related diseases. An improper diet also contributes considerably to the development of a group of diseases known as metabolic syndromes, which is expected to increase in several countries as populations age and obesity rates rise. There are various dietary risk factors associated with the development of illness; however, high salt consumption, a lack of whole grains and a lack of fruit are considered the most relevant dietary risk factors [1,2,3,4,5]. On the other hand, the world population is growing to a level where the current system of food production is not capable of regularly providing food, and it is estimated that the demand for food will increase by at least 70% of the current food production. Consequently, intensive agriculture has contributed to climate change and a considerable impact on the environment, so research is needed to develop new and sustainable food sources [6]. 

Despite the increase in the incidence of new diseases and the rapid growth of food-related illnesses, it is necessary to check the new or “old-new” sources of raw materials to promote healthier eating, not only in raw foods but also in processed foods. With the easy availability of highly processed foods, the average quality of the diet in most sophisticated market economies is low. So, one hypothesis is to change the way processed foods are made and which ingredients are used so that they are healthier solutions. Other solutions, such as changing eating habits, do not seem to have been successful enough, so effective techniques are needed to change the health problems related to poor nutrition and diet [7,8]. 

Macroalgae are one of the most promising sources of sustainable food raw materials. They can grow rapidly in water, which simplifies the manufacturing process and reduces production costs. Although macroalgae have been used for millennia in Eastern countries such as China, Japan and Korea as raw or processed food, their introduction as a food source in Western society occurred only a few decades ago, when health was declining and food-related diseases were on the rise. To combat this dietary deficiency, algae play an important role as they are rich in bioactive compounds such as polysaccharides (e.g., alginate, fucoidan, agar and carrageenan); proteins (such as phycobiliproteins); carotenoids (beta-carotene and fucoxanthin); phenolic compounds (phlorotannins); vitamins and minerals, which have also been studied against diseases such as hyperglycemia, diabetes, obesity, metabolic disorders and neurodegenerative and cardiovascular diseases [9,10,11,12,13]. 

A high level of public knowledge about a balanced diet and nutrition is essential to reduce the burden of diet-related diseases. There is, however, a dearth of scientific evidence on the public’s knowledge of diet-related diseases and risk factors. Considering the aforementioned problems, this study intends to assess how macroalgae may be utilized as a dietary supplement or as an ingredient in food, as well as to define the bioactive substances they contain that have advantageous effects on human health. In this approach, the potential benefits of macroalgae-based products may be observed in both prevalent disorders like celiac disease and more common conditions like diabetes and problems related to high cholesterol [11].

## 2. Diet-Related Diseases

### 2.1. Celiac Disease

In recent years, the number of people with diseases related to the consumption of wheat and cereals containing gluten has been increasing. The most common diseases are celiac disease, non-celiac gluten sensitivity (NCGS), wheat allergy or irritable bowel syndrome [14]. Celiac disease is a specific immune response triggered by gluten that affects the small intestine; however, individuals are also genetically susceptible to developing it [15,16]. Gluten is a general term for alcohol-soluble proteins present in cereals such as wheat, rye, barley and other derived products [17,18] and is mainly composed of gliadin and glutenin [16]. The high proline content in gluten makes it difficult for human gastrointestinal enzymes to digest it [15]. 

Wheat allergy is caused by the consumption of insoluble gliadins, which are present in wheat and can react with immunoglobulin E (IgE). One of the differences between the two conditions is that celiac disease causes permanent damage to the gastrointestinal system, and the only treatment available is to introduce a strict diet without the intake of gluten or wheat [17]. Foods such as conventional wheat bread are staple foods and are present in many people’s diets due to their nutritional value and low price [19], so eliminating this food is not one of the goals [14].

In the bakery industry, attempts have been made to produce gluten-free products that are equally flavorful and high-quality; however, some challenges need to be overcome, namely, the characteristics that gluten imparts to bread doughs and the low protein content that these foods have [14]. In conventional bread production, starch binds water and creates a gas-permeable structure. Gluten traps carbon dioxide (CO_2_), produced in the fermentation process by yeasts, which causes the dough to rise due to the presence of a minor component of gluten, the glutenin macropolymer (GMP), which imparts elastic properties to bread dough and allows it to rise [14]. Flours used as a substitute for wheat flour are rice and/or maize flours combined with starch; however, this type of mixture has a high amount of fat and salt and a low amount of protein, which affects the structure and quality of the bread [14]. Rice flour is widely used in gluten-free baking as it is a low-cost ingredient, and its characteristics, such as its white color, mild taste and easy digestibility, are suitable for bakery products [20]. However, despite the advantages described above, the functional qualities that this flour confers are not sufficient to obtain quality end products, and therefore, compounds such as hydrocolloids are added [20]. It is also possible to use pseudocereals, which are used in the production of gluten-free products due to their nutritional value, high protein content, essential amino acids and fatty acids [21]. However, it is necessary to pay attention to the growing conditions of the plant species because it influences the composition of the cereals and the quality of the final product. Legume flours, such as chickpea, soya bean and cassava, are also an alternative to starch additives due to the water retention capacity they confer [21].

Proteins are basic substances for cells and are an essential nutrient, mainly obtained through the intake of animal or vegetable protein and soya products. Milk proteins are used because they have a high nutritional value and similar chemical structure to gluten proteins; however, they can be a limited and expensive resource [21,22]. The positive side of adding milk proteins is the availability of essential amino acids they contain. On the other hand, the addition of protein-rich and lactose-poor products, which is not always the case with milk, has been shown to increase the quality of the final product, and associations have been made with lactose intolerance in people with small bowel inflammation due to celiac disease [21]. 

### 2.2. Diabetes Mellitus (DM)

Diabetes disease has also seen a large increase in recent years, and there are more and more people with impaired glucose tolerance and an increased risk of diabetes. About 537 million people have diabetes, and about 316 million people have impaired glucose intolerance and a risk of developing diabetes [23]. 

Diabetes mellitus is a metabolic disorder, determined by chronic hyperglycemia with the disorder of carbohydrate, fat and protein metabolism. As a result, there is a lack of insulin, the efficacy of its action or there is tissue insensitivity to the hormone [24]. Insulin is a hormone produced by pancreatic β-cells, and its mode of action is involved in blood glucose control, which allows cells and tissues to convert glucose into energy. If insulin or its action on tissues and cells is absent, it results in a build-up of glucose in the blood and the development of diabetes symptoms [24]. Symptoms are hardly realized because they can be attributed to other reasons; however, some of the symptoms of the most common ones can be weight loss, excessive thirst (polydipsia), excessive urination (polyuria) and dehydration, general fatigue, excessive hunger, problems with vision or blurred vision and vaginal infection. There are several types of diabetes, namely, type 1 diabetes, type 2 diabetes, gestational diabetes mellitus and other types of diabetes associated with other diseases such as pancreatic diseases, monogenic diabetic syndrome and chemical inducers [24]. Type 1 diabetes (Figure 1) is characterized by a lack of insulin secretion because of pancreatic β-cell degeneration. The body is then forced to utilize fats as an energy source instead of glucose, and there is a production of a toxic by-product, ketones. At the treatment level, patients require daily doses of insulin [24]. Type 2 diabetes (Figure 1) develops because of the progressive loss of insulin-secreting cells or tissue resistance to the hormone, where tissues do not absorb insulin and its action in the body does not occur. This type of diabetes is associated with increased body mass, which is a result of unhealthy eating patterns and lack of exercise [24].

Type 2 diabetes mellitus (DMT2) is the most common form and accounts for approximately 90% of all diabetes cases [25], being associated with other types of diseases such as high blood pressure (hypertension), chronic high insulin levels (hyperinsulinemia) and abnormal levels of cholesterol, triglycerides and lipids (hyperlipidemia). In addition, abnormalities in lipoproteins are associated with type 2 diabetes, as well as its diagnosis and treatment [24]. 

The treatment of type 2 diabetes involves stimulating and increasing endogenous insulin production or inhibiting the digestive enzymes α-amylase and α-glucosidase. α-amylase hydrolyses starch at the internal α-1,4-glucosidic bonds and produces linear or branched malt oligosaccharides. The α-glucosidase converts the previously produced oligosaccharides into glucose [24]. In this way, the two enzymes play different but interdependent roles for glucose production to occur, where the first enzyme regulates the rate of starch digestion, and the second enzyme acts in the conversion of sugars. Some synthetic inhibitors of these enzymes are already used, such as sulfonylureas, biguanides and glucose-lowering inhibitors; however, their continuous use is not advised due to problems of flatulence, cramps, vomiting, weight gain and liver function disorders [24]. On the other hand, techniques based on mechanisms of action are also used, such as stimulating insulin production and release, increasing glucose transport activity and decreasing glucose absorption in the gut. However, these types of treatments have low efficacy and side effects [26]. 

The inclusion of dietary fiber in the diet is associated with benefits in the prevention and reduction of bowel disorders, coronary heart disease and type 2 diabetes [27]. Dietary fibers are classified as non-degraded and non-digestible material in the human body and small intestine. They consist of carbohydrate polymers, present in plant cell walls, such as cellulose, hemicellulose and pectin but also polysaccharides, such as agar, carrageenan and alginate, extracted from macroalgae. Classified according to solubility, two types are distinguished, namely soluble fibers and insoluble fibers; as the names imply, the former dissolves in water and the latter does not. These carbohydrates are found in different foods and have different properties, and increasingly, their beneficial effects have been studied, such as cholesterol reduction, diabetes control with a reduction in blood glucose levels and better functioning of the digestive system [27]. In addition, this type of fiber also improves the growth and activity of intestinal bacteria, exerting a prebiotic activity.

Diabetes, as both acute and chronic hyperglycemia, is associated with the production of reactive oxygen species (ROS), which activates cell apoptosis. The induction of ROS production is through mitochondrial enzymes of the respiratory chain, namely xanthine oxidases, lipoxygenases, cyclooxygenases, nitric oxide synthases and peroxidases [28]. In different studies associated with diabetes, the imbalance between oxidizing species has already been studied, which, as a result, caused oxidative stress and cell death. Therefore, the down-regulation of ROS production may be important in controlling diabetes-associated complications [28,29,30]. Oxidative stress is defined as the imbalance in the production and neutralization of reactive oxygen and nitrogen species [24], and to maintain cellular homeostasis, cells need to balance ROS production and consumption. 

Antioxidants are molecules that slow down or prevent oxidation processes, and as a result, there is neutralization and elimination of ROS in the body’s cells. These compounds can be obtained from the diet, but there are also commercial synthetic antioxidants, such as butylated hydroxytoluene (BHT), propyl gallate (PG) and butylated hydroxyanisole (BHA); however, they are unstable, can cause side effects and are associated with promoters of carcinogenesis [24,31]. 

### 2.3. Cholesterol

The circulation of lipids in the bloodstream, such as cholesterol, occurs through lipoproteins, such as low-density lipoprotein (LDL) and high-density lipoprotein (HDL). Pancreatic β-cells can synthesize cholesterol via the mevalonate pathway, and its excess is removed by the reverse cholesterol transport process [32]. There are associations between cholesterol accumulation and pancreatic β-cell dysfunction, with individuals with type 2 diabetes exhibiting lipid abnormalities, which contribute to cholesterol accumulation in the cells and influence insulin secretion [32,33]. 

Dyslipidemia is characterized by the presence of high levels of lipids in the blood and is usually associated with abnormal glucose metabolism. Both normal lipid parameters, such as triglycerides (TG), total cholesterol (TC), high-density lipoprotein cholesterol (HDL-C) and low-density lipoprotein cholesterol (LDL-C), as well as non-traditional parameters, are related to the development of diabetes. Glycemic control is beneficial for lipoprotein levels in diabetes as there is a reduction in cholesterol and triglycerides through a decrease in LDL and an increase in LDL catabolism [34]. 

## 3. Macroalgae

Currently, more than 70 species of algae have been approved for food consumption, with each type having a varied nutrient composition according to the species and abiotic factors such as location, temperature and habitat [35]; moreover, no species is harmful to human health [36]. 

Marine macroalgae are aquatic, multicellular, macroscopic organisms that, based on thallus color, pigment composition and biochemical aspects, can be classified into three groups: Chlorophyta (green algae), Rhodophyta (red algae) and Ochrophyta—Phaeophyceae (brown algae) [37,38]. In certain aquatic ecosystems, algae act as primary producers as they synthesize organic material and oxygen for the metabolism of consumer organisms [39]. As they are species that live in extreme environments and with variations in the different factors, they need a rapid readaptation to the changes that occur, producing primary metabolites (proteins, amino acids, polysaccharides and fatty acids) that act as stress deterrents and secondary metabolites (phenolic compounds, pigments, vitamins and other bioactive compounds) as a response to the changes that occur in the environment [40]. 

Although more research and discovery are still needed on macroalgae, it is already known that the substances that these beings synthesize have great potential in different areas such as pharmaceuticals, food and cosmetics. In the pharmaceutical and cosmetic industry, macroalgae are useful as they are used as a source of natural ingredients or bioactive compounds. They are also used in human food, which is quite prevalent in Asian countries, but they are also used in livestock and agriculture, animal feed, nutritional additives and stimulants for plant growth [38]. 

The demand and need for the consumption of non-toxic and environmentally friendly products has increased. Therefore, algae have started being used as a source of natural compounds and as a replacement for synthetic food additives [38,41]. Compared to other taxonomic groups, macroalgae are rich in biologically active compounds such as polysaccharides, phenolic compounds, proteins, polyphenols and pigments, as well as certain micronutrients such as potassium, sodium, calcium and iron [37]. There is also an interest in nutraceutical preparations as a health benefit, and macroalgae have potential as dietary supplements and their consumption is associated with beneficial health effects due to the components they contain. Natural bioactive chemical substances known as nutraceuticals—also known as phytochemicals or functional foods—have qualities that can improve health, stave off disease or treat certain conditions [42]. 

Nutraceutical products are non-specific biological medicines that are used to increase overall well-being, manage symptoms and prevent cancer. The phrase “nutraceutical” is a combination of the terms “nutrient”, which refers to a nutritious dietary component, and “pharmaceutical”, which refers to a medicinal medicine. Stephen DeFelice, the founder and chairman of the Foundation for Innovation in Medicine, an American organization based in Cranford, New Jersey, invented the term in 1989. The definition of nutraceuticals and associated items varies according to the source. These goods can be classed according to their natural origins, pharmacological conditions, and chemical composition. Nutraceuticals are often classified into four categories: dietary supplements, functional foods, medical foods and pharmaceuticals [43]. 

Bioactive molecules that can be incorporated as a supplement in the diet can be considered nutraceutical food products [44]. The in vitro biological activity of several compounds with antioxidant capacity has already been studied [45,46], as have their anti-inflammatory properties [47], applications in the treatment of Alzheimer’s and Parkinson’s disease [48], applications in the control of glycemic index [49] and potential as antidiabetics [50].

In 2015, a set of 193 countries came together to create a plan to improve the health of the planet; they listed 17 Sustainable Development Goals (SDGs), all interlinked with each other, aiming for a better and more sustainable future. These goals address issues such as poverty, inequality, climate change, environmental degradation, peace and justice [51]. Macroalgae have the potential to address these sustainability challenges as they are a source of food (SDG 2), can provide clean water (SDG 6), green energy (SDG 7), improve global health (SDG 3) and are a renewable and fast-growing source of biomass [51]. 

According to the problems mentioned above, macroalgae have the potential to be used in the production of certain foods due to their high content of proteins, amino acids, and minerals, which have been shown to increase the nutritional value of different products [52]. On the other hand, the antidiabetic effect, glycemic index control and cholesterol control have been demonstrated through compounds present in macroalgae, making it possible to develop products with potential for the treatment or improvement of diseases.

Seaweed bioactive chemicals have the potential to have a significant therapeutic role in disease prevention in humans. Seaweed bioactives such as polysaccharides, pigments, fatty acids, polyphenols and peptides have been shown to have a variety of positive biological qualities that might possibly contribute to the creation of functional foods and nutraceuticals [53]. In recent years, seaweed has grown in popularity as a more adaptable culinary item that may be used directly or indirectly in the manufacture of dishes and drinks. Seaweed and its products are particularly important in the food business due to their utility as components in fertilizers, animal feed supplements and additives for functional meals.

Furthermore, functional food and nutraceutical products based on seaweed constituents should be evaluated for the presence of pollutants, allergies, heavy metals or dangerous chemicals created during seaweed farming or processing. To move on with commercial development and manufacturing, these items must comply with and adhere to strict safety standards [53].

### 3.1. Proteins 

Proteins are macromolecules that perform various functions in the body, such as building bones, muscles, cartilage, skin and blood, and they are precursors to other molecules, such as enzymes, antibodies and hormones. Despite their importance, the body does not have the capacity to store proteins and, therefore, needs supplies from other sources. The addition of proteins from different sources, such as animal, vegetable, and soy products, is widely used in products with a low content of these compounds, but it is an expensive process, and there are links between the consumption of this type of protein and certain intestinal diseases [20]. Macroalgae can, therefore, be a substitute for these sources, as these proteins are also found in the composition of these living beings, such as the macroalga *Saccorhiza polyschides* (Ochrophyta, Phaeophyceae) (Figure 2a) (14%) and species of the genus *Gracilaria* (Rhodophyta) (Figure 2b) [52,54]. Seaweed is considered to be a nutraceutical food since it has higher protein values than legumes and soybeans and is, therefore, becoming increasingly common. In addition, this type of compound present in red algae has demonstrated bioactive properties with pharmaceutical potential. The percentage of protein in macroalgae varies according to species, and it is known that this parameter depends on the bioavailability of nitrogen in the water, being higher during the winter and early spring [11]. 

The high levels of phenolic compounds, such as tannins, and certain polysaccharides are important and concentrated in the digestibility of algae proteins, as they have the ability to bind to proteins and form insoluble complexes; however, this phenomenon can be overcome through the amount of heating of seaweed, which causes partial denaturation and allows the breakdown of proteins into smaller peptides that are easily digestible [11]

### 3.2. Polysaccharides from Macroalgae

Hydrocolloids or polysaccharides can be used as gluten substitutes due to the stabilization and texture enhancement that they provide [21]. These compounds are used as thickening agents and stabilizers to improve the quality of the final product and to extend its shelf life [21,55]. These types of molecules are part of the composition of macroalgae and are used for various purposes in food products, such as water retention in meat products; increasing viscosity in soups, broths and beverages and as gluten substitutes [56]. The addition of algae to gluten-free bakery products favors the polysaccharide network. In studies where different concentrations of macroalgae were added to the dough of gluten-free products, the authors report that the algae interacted with other components present in the dough and contributed to an improvement in the rheological properties of the dough and a greater retention of gas (CO_2_) that caused an increase in the volume of the dough and the final product [52]. Therefore, algae are good substitutes for gluten in the bakery industry.

Brown algae (Ochrophyta, Phaeophyceae) contain alginate and fucoidan, while red algae (Rhodophyta) contain polysaccharides such as agar and carrageenan. Finally, green algae (Chlorophyta) contain cellulose and hemicellulose [57].

Alginate (Figure 3) is extracted from brown macroalgae and consists of linear, unbranched compounds of β-1,4-D-mannuronic acid and α-1,4-ʟ-guluronic acid. These types of polysaccharides are classified as phycocolloids due to their ability to form colloidal systems in the presence of water. In the presence of metal ions such as calcium, sodium or magnesium, alginic acid reacts with the ions and forms alginate. Calcium alginates are insoluble in water, but with sodium and magnesium ions, these alginates become water-soluble [58].

Alginates are used industrially due to their stabilizing, thickening and emulsifying characteristics, but due to more specific properties, namely gel strength, porosity and biocompatibility, they have been increasingly used as biomaterials in engineering [58].

Fucoidan (Figure 4) is a polysaccharide sulfate synthesized by brown algae. It mainly contains fucose but may have other types of monomers such as glucose, galactose, xylose and mannose. Depending on the species of macroalgae from which they are extracted, fucoidans may have different lengths of chain, branching structures and sulfate compound contents. This type of polysaccharide can be divided into two subgroups: one group with residues of α-1,3-fucopyranose and a second group with alternating residuals of α-fucopyranose at 1,3 and 1,4 [58].

Fucoidans are important in regulating water retention and ions in algae cell walls to avoid water loss and osmotic stress in the presence of low-tide seasons [58].

Agar (Figure 5) is a sulfate polysaccharide present in the cell walls of red macroalgae, such as the genus *Gelidium* and *Gracilaria*. It consists of agarose, a neutral polysaccharide with a linear and repeated structure of agarobiose disaccharides, and agaropectin, an acidic polysaccharide composed of sulfate, methyl, pyruvic acid and glucuronic acid [59].

Carrageenan (Figure 6) is a sulfate polysaccharide, also present in the cell walls of red macroalgae. It has a high molecular weight and is composed of α-1,3-d-galactopyranosyl and β-1,4-d-glucopyranosyl groups bound to residues of 3,6-anhydrogalactose [60]. The three most important types of carrageenan are kappa (κ), iota (ι) and lambda (λ), which differ in the position of ester sulfate groups and the number of anhydrogalactose. Carrageenan extracted from *Chondrus crispus* (Rhodophyta) can be used as a gelling and thickening agent in gluten-free bakery products. It helps improve the texture and consistency of gluten-free doughs and batters, addressing the challenges associated with gluten replacement while enhancing the nutritional value of the end products [61].

As mentioned earlier, algae produce different polysaccharides, which have already demonstrated their antidiabetic effects. For example, red algae contain sulfate polysaccharides, such as agar and carrageenan, which in various studies [62,63,64] have already verified their ability to inhibit digestive enzymes, namely α-amylase and α-glucosidase, and as a result, reduce blood glucose levels. The antioxidant effect of polysaccharides extracted from macroalgae, which can contribute to the removal of reactive species of oxygen associated with diabetes, has also been studied [25]. Hyperglycemia that occurs in diabetes promotes the production of oxidative species of oxygen (ROS), contributing to oxidational stress and inflammation. Thus, natural compounds present in macroalgae have important biological activities in neutralizing ROS, helping in the control of cellular damage, as well as in other processes involved in diabetes [25]. This event is quite important in controlling blood glucose peaks after meals and food intake. Specifically, in one study, the inhibition of digestive enzymes with *Fucus vesiculosus* (Ochrophyta, Phaeophyceae) (Figure 7) extracts was demonstrated in comparison with the commercial drug acarbose [65].

There is a beneficial effect of the consumption of dietary fibers with certain diseases such as coronary heart disease, type 2 diabetes and some intestinal disorders [27]. In their composition, macroalgae contain these types of compounds, both soluble and insoluble fibers, which can slow down the digestion and absorption of glucose [26]. In addition, fiber is capable of interesting biological activities, such as lowering cholesterol and blood glucose levels [27]. In one study where obese diabetic individuals ate dried algae, a decrease in glucose concentration and a beneficial change in lipid levels was observed [66].

### 3.3. Phenolic Compounds

In several secondary metabolites, macroalgae contain high concentrations of phenolic compounds, which are essential for defense mechanisms against biotic and abiotic stress factors. Phenolic compounds are made up of a benzene ring structure that binds to hydroxyl groups [67], which removes metal ions, such as Cu and Fe, through chelation processes [68] with a wide variety of chemical structures and different types, such as phlorotannins, bromophenols, flavonoids and amino acids similar to mycosporine [69].

This type of molecule has different biological activities, such as antidiabetic, anti-inflammatory, antimicrobial and antioxidant activities, and its activity is related to the interaction between proteins. In several studies, the antioxidant activity of macroalgae extracts has been tested, obtaining promising results and a great potential for ROS removal [64,67,70,71].

Extracts rich in phenolic compounds have beneficial biological activities for the food industry. In the literature, their ability to increase the shelf life of food products due to their antioxidant and antimicrobial properties has been described [11,72].

Within this class, the phlorotannin dieckol is important in the food industry and is already considered a novel food by the EFSA. Dieckol is a phlorotannin found in brown algae of the *Ecklonia* species and is currently used as a food supplement and is considered safe according to the daily intake limit, depending on the age of the consumer [38]. Phlorotannins present in brown algae, such as *Ascophyllum nodosum*, *Fucus distichus* and *Padina pavonica*, have demonstrated antidiabetic properties [72].

Flavonoid compounds have been studied as a possible antidiabetic treatment in a study with *Ulva prolifera*, wherein flavonoid-rich extracts promoted a decrease in fasting blood glucose, which increased oral glucose tolerance and demonstrated modulation of the gut microbiome that positively influenced insulin release and resistance [73]. Of the phenolic compounds present in red algae, bromophenols have shown interesting results as antidiabetic and anti-obesity agents. Extracts of *S. latiuscula* with bromophenols inhibited α-glucosidase and improved insulin sensitivity and glucose uptake [74]. Bromophenol derivatives extracted from *Rhodomela confervoides* demonstrated activity against protein tyrosine phosphatase 1B (PTP1B), a negative regulator of the insulin signaling pathway, which caused a considerable decrease in the blood glucose of diabetic rats [75].

As a food application, this type of compound can be used as a natural food stabilizer and preservative. The restrictions that exist around synthetic compounds used in the food industry can be the starting point for the investigation and use of seaweed compounds as an alternative to existing synthetic compounds, as they have antimicrobial activities that help prevent the deterioration of food, and the development of pathogenic microorganisms. On the other hand, they are also used as oxidative stabilizers to preserve and increase the quality and nutritional value of foods [72].

### 3.4. Pigments

Pigments are fat-soluble polyenes that provide color but are also capable of biological activity. Macroalgae are rich in pigments such as chlorophylls, carotenoids and phycobilins. Depending on the type of algae, different types of pigments can be found, such as chlorophylls, present in green macroalgae (Chlorophyta); phycocyanin and carotenes, present in red algae (Rhodophyta) and, finally, β-carotene, fucoxanthin and violaxanthin, which are present in brown algae (Phaeophyceae) [11,25,76].

The consumption of natural pigments is associated with human health, and there are different types of potential functional ingredients. However, in this context, the one that has gained prominence in the food industry is fucoxanthin. The interest in this pigment lies in its antioxidant properties, but it is also known to have stimulated glucose absorption in cells and improved insulin sensitivity. This type of molecule activates the signaling pathways involved in the production and action of insulin, such as the PI3K/Akt pathway, which promotes the absorption of glucose in the cells, contributing to the control of the glycemic index. Currently, fucoxanthin derived from *Phaeodactylum tricornutum* (Bacillariophyta) is authorized by the FDA (United States) and the Food for Specified Health Uses (Japan), and there are no records of toxicity and adverse effects on human health [11,25].

### 3.5. Carotenoids

Carotenoids are natural pigments found in plants, fungi, algae and bacteria. These molecules are not synthesized by the body, so they need to be ingested via diet or supplements. Different carotenoids already exist, such as β-carotene, lutein, lycopene and canthaxanthin. Another use for carotenoids is as colorants in food, beverage and pharmaceutical applications [77].

Among the different functions that carotenoids have in contributing to human health, β-carotene stands out due to its ability to be converted into vitamin A, while lutein helps protect the eyes due to its absorption of light at specific wavelengths. Lycopene, another carotenoid available in macroalgae, has demonstrated antioxidant activity in vitro and reduced serum cholesterol in animal studies [77]. These compounds have also been shown to act on various mechanisms, leading to a reduction in abdominal and subcutaneous fat in an in vitro study [78]. The antioxidant power of carotenoids is also important, and a reduction in overall oxidative load and a decrease in its accumulation has been found, resulting in weight and obesity control [79].

In one study where a formulation for whole meal cookies was developed, astaxanthin was added, and there was a reduction in glucose during in vitro digestion and an increase in phenolic content and, consequently, antioxidant activity compared to control cookies [80].

In a clinical trial involving 17 children with obesity, they reported the effectiveness of supplementation with mixed carotenoids on obesity and markers of insulin resistance. For obesity, they found an improvement after 6 months of supplementation with these pigments [81].

### 3.6. Fatty Acids

Macroalgae contain low concentrations of lipids (2–10% of their dry weight), the most common being phospholipids and glycolipids. Most of the lipids present in algae are of the polyunsaturated type (PUFAs), and the concentration varies according to the region where they are found. PUFAs can be classified into omega-3 and omega-6 fatty acids, and the most important include essential fatty acids (EFAs) such as eicosapentaenoic acid (EPA) and docosahexaenoic acid (DHA). EPA is found in various macroalgae, mainly red macroalgae, for example, *Palmaria palmata*, where this fatty acid accounts for up to 50% of the total fatty acid content. On the other hand, certain products of the synthesis of PUFAs are also present in some edible macroalgae; however, humans cannot obtain these compounds in the quantities required, so looking for other sources, such as food, is important for human health as these molecules are essential for a balanced metabolism and are, therefore, important ingredients in the diet [76,82].

### 3.7. Mycosporyne Amino Acids

Macroalgae, due to environmental factors such as ultraviolet radiation, pH, temperature and salinity, have developed defense systems such as DNA repair and cell protection through the production of various compounds such as mycosporine-like amino acids (MAAs). MAAs have different biological activities, including anti-UV, anti-inflammatory, anti-cancer and anti-aging activities, but they also contribute to stimulating cell proliferation, wound-healing and DNA protection [83]. Red algae have the highest concentration of these compounds and can be found in various species such as *Asparagopsis armata*, *Chondrus crispus*, *Mastocarpus stellatus*, *Palmaria palmata*, *Gelidium* spp., *Pyropia* spp. (formerly known as *Porphyra* spp.), *Crassiphycus corneus* (formerly known as *Gracilaria cornea*), *Solieria chordalis*, *Grateloupia lanceola* and *Curdiea racovitzae* (Rhodophyta) [72].

The fact that this type of molecule has antioxidant effects can help reduce oxidative stress in the body, which is linked to various diseases, including diabetes, cardiovascular disease and Alzheimer’s disease [84].

There are different MAAs with slight differences in their structure and chemical and physical properties. MAA precursors include gadusol, which has gained importance due to its antioxidant capacity, verified in studies where the relative oxygen radical absorbance capacity (ORAC) was evaluated in comparison with flavonoid antioxidants such as ascorbic acid, quercetin and rutin, obtaining promising results. Another type of MAA is mono-substituted MAAs, and the antioxidant capacity of certain MAAs such as mycosporine–glycine (M-Gly), mycosporine–taurine (M-Tau) and mycosporine–γ-aminobutyric acid (M-GABA) has also been studied. Finally, there are two more types of MAAs, di-substituted and derivatized, which have also demonstrated antioxidant characteristics. Interestingly, the first type of MAAs mentioned, the di-substituted ones, can be found in marine algae such as the red alga *Porphyra tenera,* where the MAA Porphyra-334 was detected, and the alga *Chondrus yondoi*, where palythine was detected, and the radical-scavenging capacity of extracts from these algae was verified [85]. In another study, which carried out in vitro tests with aqueous seaweed extracts, the presence of Porphyra-334, shinorine, asterin-330 and palythine extracted from the algae *Porphyra rosengurttii*, *Gelidium corneum* and *Ahnfeltiopsis devoniensis*, respectively, was determined, confirming the ability of these compounds to eliminate reactive oxygen species [86].

Due to the characteristics of MAAs, the use of these compounds for other purposes has been studied. MAAs extracted from the red alga *Porphyra rosengurttii* acted as modulators of signaling pathways, particularly in pathways associated with inflammation and oxidative stress, and are, therefore, potential candidates in the pharmaceutical field. On the other hand, a cream was produced with the alga *Porphyra umbilicalis*, which helped protect against UV radiation, improved skin smoothness and firmness and reduced the depth of wrinkles and lipid peroxidation. In other studies looking into the use of MAAs as sunscreens, promising results have been obtained with compounds extracted from red algae (*Porphyra rosengurttii*, *Gelidium corneum* and *Ahnfeltiopsis devoniensis*) and microalgae (*Chlamydomonas hedleyi*), proving their importance in the cosmetics and pharmaceutical industry, as they are natural and safe and protect against various effects caused by solar radiation [87,88].

### 3.8. Cholesterol

Cholesterol is important to cell structure and enters various metabolic pathways, and regulation of this compound is important for normal lipid metabolism. Deregulation of cholesterol in the body is associated with various metabolic diseases, such as type 2 diabetes, heart disease and liver disease [89]. Phytochemicals have beneficial effects in reducing cholesterol levels by inhibiting the expression of enzymes and transcription factors associated with metabolism. Phytochemicals are secondary metabolites synthesized naturally in microalgae or macroalgae (for example, [89]) and include carotenoids, sterols and polysaccharides. In some studies, they used hazardous algae extracts in phytochemicals as a dietary supplement and observed a reduction in total cholesterol, triglycerides and LDL levels [90,91,92]. On the other hand, antioxidants compounds present in macroalgae, such as polysaccharides, fucoxanthin and phenolic compounds, have also been shown to reduce cholesterol levels [93,94,95,96,97].

## 4. Algae Food Products as Healthcare Solution

Algae demonstrate a potential for use in food products, such as bread, and their effects have been studied in several studies. The fact that they have a high content of minerals, proteins and amino acids can provide a higher nutritional value compared to conventional foods [51]. On the other hand, there is increasing interest in nutraceutical preparations or food supplements as a health benefit, and macroalgae have great potential for this purpose due to their components with beneficial properties for health [44].

### 4.1. Red Algae (Rhodophyta)

*Porphyra/Pyropia* (Rhodophyta) (Figure 8), commonly known as Nori, is a red alga rich in protein, vitamins and minerals. It can be incorporated into gluten-free snacks or crackers, providing a source of essential nutrients for individuals with celiac disease while also offering a low-carbohydrate option for those managing diabetes or hyperglycemia [98].

*Palmaria palmata* (Rhodophyta) (Figure 9), also known as Dulse, is a red algae species with antioxidant properties. Its powdered form can be included in gluten-free baking mixes, offering both nutritional benefits and contributing to the control of oxidative stress associated with diabetes [22]. On the other hand, these red algae contain high levels of protein, between 9 and 25%, which in bread production has been shown to increase the nutritional profile of bread, which is accepted by consumers and has the potential to be developed as a bioactive product [99].

Certain species of *Gelidium* (Rhodophyta) can produce extracts rich in bioactive compounds, including antidiabetic agents. These extracts can be incorporated into functional beverages or health supplements targeting glycemic index control and diabetes management [100].

*Gracilaria* species have been studied for their ability to reduce cholesterol levels. Incorporating *Gracilaria* extracts into food products, such as gluten-free pasta or spreads, could help support individuals with celiac disease and diabetes while addressing cholesterol-related health concerns [98,101]. The genus *Gracilaria* contains more than 300 species, and 160 are taxonomically accepted [102]. It is of great economic importance as it has high biomass yields and the ability to produce agar. The species of the genus *Gracilaria* produce weaker gels compared to other macroalgae genera; however, they are considered one of the most important sources of agar because they are fast-growing algae with a relatively low acquisition cost [38,102]. The addition of polysaccharides extracted from this type of algae has been shown to improve the nutritional, structural and shelf-life characteristics of foods. These species have low lipid contents and contain docosahexaenoic acid (DHA), a polyunsaturated fatty acid that is important in reducing the risk of cardiovascular diseases. In terms of amino acids, the most abundant are aspartic acid, alanine, glutamic acid and glutamine. Still, they are also a good source of soluble and insoluble dietary fibers and are, therefore, good alternatives to cereal-based fibers, which are used in Western countries [54].

Allsopp et al. [103] enriched bread with red algae and evaluated, in a study with humans, how its intake can influence markers of inflammation, lipid profile, thyroid function and antioxidant capacity.

### 4.2. Green Algae (Chlorophyta)

Green algae (Figure 10) contain a variety of compounds with interesting biological activities. Species such as *Ulva clathrata*, *Ulva compressa*, *Ulva intestinalis*, *Ulva linza* and *Ulva flexuosa* have been found to have a high capacity to eliminate radicals and can be used in pharmaceutical, cosmetic and food products [69].

Flavonoids have been studied as antidiabetics, and extracts rich in this compound have been found to lower blood glucose and positively influence insulin release and resistance [69].

Species belonging to the *Ulva* and *Monostroma* genera are of great interest as food due to their high protein content and polyunsaturated fatty acids [36].

The use of green macroalgae in food or as a food supplement is still limited, and there are few studies in the literature. However, it should be noted that this type of algae also has great potential for food use due to the varied compounds it contains with bioactive and health-benefiting properties.

### 4.3. Brown Algae (Phaeophyceae)

Brown macroalgae show a high concentration of secondary metabolites, which have shown antidiabetic activity in vivo, and studies have seen a reduction in blood glucose, triglycerides and cholesterol levels, as well as hepatoprotective activity [104,105]. In studies with diabetic rats, a decrease in glucose levels, weight loss, and normalization of triglyceride and cholesterol levels were observed. A healing effect and regeneration of pancreatic cells were also observed in induced diabetic rats after administration with different concentrations of algae extracts [106,107].

*Saccorhiza polyschides*, initially classified as *Fucus polyschides*, is considered an edible species and has, therefore, been studied for future nutraceutical applications [108]. It is a brown alga (Phaeophyceae) belonging to the family Phyllariaceae and the class Phaeophyceae [52,109]. This alga is opportunistic and is most abundant in southern Europe, where it colonizes rocky substrates in the sublittoral [52]. Its sporophyte is pale dark brown, with differentiated blades and a twisted base, and the attachment zone is composed of a rough “bulb”. It is a monocarpic and annual alga, but it is fast growing and can reach between 2 and 4 m in length in just 2 months [52,109]. It is a species that contains a high percentage of carbohydrates (45.6%) and protein (14.4%) as well as low levels of lipids (1.1%), and it has a concentration of minerals higher than that found in terrestrial plants. Due to its ability to adapt to environmental changes, such as salinity and temperature, it contains bioactive compounds synthesized in the presence of these types of changes [52].

Różyło et al. [52] evaluated the influence of adding brown macroalgae on the properties of gluten-free bread, namely, the physical and sensory properties and the antioxidant capacity. They obtained promising results in the production of this type of bread.

Studies in which edible Wakame seaweed (*U. pinnatifida*) was added to pasta in different percentages found that it was accepted and had a slight taste of seaweed but was similar to the control sample. It was also found that the addition of the algae improved the quality of the pasta, as there was an interaction between the starch and the protein matrix [110].

Beyond enhancing nutritional profiles, *A. nodosum* and *F. vesiculosus* offer a multifaceted approach to managing celiac disease and diabetes. Their bioactive compounds, such as sulfated polysaccharides, have demonstrated impressive antidiabetic effects, which include controlling glycemic index levels and mitigating the adverse effects of hyperglycemia. Moreover, these seaweed-derived compounds show promise in effectively regulating cholesterol levels, thereby reducing the risk of related health complications [111].

### 4.4. The Use of Macroalgae in Gluten-Free Products

Studies addressing the effect of adding certain macroalgae to pasta or flour in bread production are still scarce. The first to conduct studies of this type were Medvedeva et al. [112], who examined how the use of algae products can increase the nutritional value of bread. Mamat et al. [113] verified the rheological properties of the bread paste and the quality of the same after adding seaweed to the flour. The beneficial effects that bioactive compounds, which are present in macroalgae, can present in different metabolic diseases are already known. The studies mentioned above mainly refer to wheat bread; however, studies where algae are added to gluten-free bread are even more scarce. Within products that do not contain gluten, gluten-free pasta has already been developed with the addition of algae to rice flour, resulting in a reduction in energy value, increased protein content as well as increased insoluble dietary fiber [113]. Studies where there is the evaluation of different factors, such as the influence of macroalgae compounds in the production of gluten-free bread and possible diabetes treatments and the reduction of cholesterol through the properties of the bioactive compound, are not found in the literature, so there are strategies and factors worth studying so that foods or nutraceutical preparations based on macroalgae can be produced.

In light of the aforementioned challenges, certain macroalgae species, such as *Ascophyllum nodosum* (Figure 11), *Fucus vesiculosus* (Phaeophyceae), *Cladophora* spp. and *Ulva* spp. (Chlorophyta) present a promising avenue for the development of specialized foods owing to their abundant reservoirs of proteins, amino acids and minerals. Studies, such as the ones conducted by Różyło et al. [52] and by Menezes et al. in 2015 [114], have indicated that incorporating these macroalgae species into food formulations can significantly elevate the nutritional value of various products. This becomes particularly crucial when addressing the limitations of gluten-free items in the bakery industry, as macroalgae-based additives may help compensate for the loss of gluten’s beneficial characteristics.

By harnessing the inherent potential of *A. nodosum* and *F. vesiculosus*, it becomes feasible to develop innovative food products that can serve as therapeutic interventions or improvements for individuals with celiac disease and diabetes. Such products may not only cater to the dietary needs of these specific populations but also provide added health benefits through the incorporation of these specific macroalgae-derived bioactive nutraceuticals. As research in this field continues to unfold, with a focus on exploring other macroalgae species as well, it is anticipated that macroalgae-based foods will emerge as a valuable adjunct to conventional treatments, contributing to a more comprehensive and effective approach to managing these diseases and improving the overall well-being of affected individuals [114].

It is important to note that the utilization of macroalgae species in food products requires careful research and formulation to ensure safety and effectiveness. Additionally, individuals with allergies or sensitivities to seafood should be cautious when consuming products containing seaweed or algae-derived ingredients [9,115].

## 5. Algae Food Industry: A Possible Key Road?

Algae are increasingly being used for functional advantages that go beyond the typical nutritional and health considerations. There is extensive evidence for the health advantages of algal-derived food items, but quantifying these benefits, as well as the potential harmful consequences, remains difficult. However, related to the abovementioned, there is still a long road ahead due to several bottlenecks that hinder the potential use of macroalgae as a key ingredient in the food industry.

### 5.1. Algae as Raw Source

Algae are renewable sources of proteins, minerals and fiber, as well as vital amino acids, pigments and fatty acids, among other metabolites important for human nutrition. Unlike the intensive cultivation that occurs in Asia, the European and American macroalgae industries are still in their early stages and have been virtually entirely reliant on wild stock collection. This is a huge bottleneck for the food industry’s incorporation of more algae into food products. Cultivation is viewed as a more sustainable approach than the overexploitation of wild seaweed resources; this is required to fulfill the processing industry’s rising need for traceable, high-quality and predictable biomass outputs. Furthermore, when cultivated in conjunction with fed and extractive aquaculture species of different trophic levels in integrated multi-trophic aquaculture (IMTA) systems, seaweed may be produced sustainably [116].

### 5.2. Algae as a New Food

Although seaweed has been consumed for many years, there are still drawbacks to its use in human food due to restrictions imposed by legislation and risks that may be associated with its consumption. In the European Union, there are already 18 species of macroalgae approved for marketing and human consumption; 3 species are accepted and authorized under the Novel Foods Regulation, and 6 species are currently being investigated for possible applications and consumption [40].

There is a growing interest in healthy foods due to the increase in diseases related to diet and the type of diet practiced. This type of food is associated with relatively expensive market prices, although the main ingredients are based on antioxidants, vitamins or minerals. Seaweed contains these types of bioactive compounds and is, therefore, a good candidate as a superfood. In addition, there is a link between the consumption of macroalgae and direct benefits for human health [56].

Seaweed is used in various industries (food, nutraceuticals, cosmetics, pharmaceuticals, etc.), but the increase in its use, as well as legislation and authorizations to allow its consumption, will depend on coming generations and develop over the following years, depending on how research and market demand evolve. On the other hand, there also needs to be an evolution in terms of the production, processing and techniques used for macroalgae to increase their efficiency. Seaweed can be incorporated into a wide variety of products; it is an innovative path for research and production to exploit the potential of these beings in different industries. Currently, there is a focus on sustainability to help the environment, and seaweed is an “agricultural” system where the associated costs are minimal since fertilizers, feed and pesticides are not as needed compared to other systems. In certain countries, aquaculture and macroalgae systems are already replacing other industries, and there has been an improvement in people’s economic situation [56].

Despite the advantages mentioned above, there are risks associated with consuming macroalgae. These beings are susceptible to toxic products that are present in the environment and are difficult to control, so there are concerns related to the absorption and incorporation of minerals, heavy metals and other compounds in seaweed, which could be transmitted through their consumption, resulting in contamination and allergies. It is difficult to control the nutritional composition of macroalgae, as well as the quality of the end product, due to the changes that occur in the environment, which directly influence the chemical composition of the product. On the other hand, it is also difficult to evaluate algae in general terms because they vary greatly between species [56].

There is a need for more information on macroalgae and the benefits associated with their consumption, as well as the positive impacts they have on the environment, to overcome the challenges of incorporating new products or foods.

### 5.3. Food Safety

The nutritional potential of algae is directly tied to their biochemical makeup and bioactive qualities, which are known to vary greatly between classes and even strains. There is a lack of awareness of nutritional content among algae species, geographical locations, and seasons, which can all have a significant impact on their dietary value. However, in recent years, the biochemical composition and functional properties of algae feedstocks have been extensively studied over the last few decades, aided greatly by the advancement and development of new techniques that enabled high-resolution profiling of proteins, lipids, polysaccharides, pigments and others. Overall, algae offer a sustainable supply of natural high-value bioactive chemicals with the potential to produce novel human nutrition products, mostly as raw sources; albeit, this is in a simplistic overview, without normal model assays of the next bottlenecks [82,116].

The impacts of harvesting, storage and food processing can all have a significant impact on the potential nutritious content of algal-derived meals. Thus, there is a need to develop chemical assays to guarantee the maintenance of the raw material quality before being applied to the food industry to certify the quality and maintenance of the raw source, boosting the confidence of the final consumer. This can be achieved using software and rapid assessment techniques such as spectrophotometry, TLC and FTIR, which can be applied to software that can give similarity rates between samples, which can be used as material pre-testing before being approved and used in the food industry [117,118,119].

Another question is during the food processing, which can modify the algae compounds and organoleptic from the algae. Thus, this also requires a high interest in promoting food safety and reducing problematic cases. Due to general analysis of the compounds in their natural state rather than the hydrolysate stage (caused by heat or pressure) with low molecular weight, the overall effects on the chemical structure of the compounds (primarily molecular weight, bioactivity and toxicity) when algae are cooked/prepared are not well known. Furthermore, understanding the kinetics of food-processed algae vs. stability may be critical for gaining important insights into the impacts of the processes, ensuring improved food safety levels and retaining a comparable product with stable chemicals that can be bioavailable to humans. Thus, it is a key point to promote a rigorous analysis of the final product and also create a certifying system to promote food safety before human consumption beyond the required minimal assays, such as nutritional profile and bacteriological assays [9].

### 5.4. Algae as a Nutritional Source

Although there is extensive evidence for algae as nutritional and functional meals, quantifying these advantages and analyzing potential detrimental consequences remain significant problems. A vital bottleneck for the use of seaweeds is determining which fractions of algal diets are accessible to humans and what factors determine how food constituents are released, which might range from food preparation to genetic differentiation in the gut microflora. Understanding how algae’s nutritional and functional elements interact with human metabolism is another question that is in the process of being answered [82]. It is impossible to overestimate the significance of determining the biological availability of nutritional and functional dietary components. Bioavailability is important for both proportionate digestion and absorption of nutrients and functional food components, as well as the degree of fermentation and the type of host-microbial co-metabolism in the colon. As a result, despite very accurate and exact food content assessments, the existing understanding of the nutritional or functional food value of algal products is essentially qualitative. To test the bioavailability of nutritional and functional components of algae utilized in all diets, adequate model systems and rigorous experimental design are required. This area needs more studies to further evolve the algae into food products with certified health benefits that are not only from a nutritional point-of-view [82].

### 5.5. Future Road

An increasing global population, resource depletion, environmental challenges and climate change necessitate a new approach to food and economic systems. This requires the development of new and sustainable methods of feeding the world’s constantly rising population. In recent times, these changes have increased to a keystone and vital importance due to numerous military and political conflicts, agricultural restrictions and healthy food rules that increase the high global energy, commodity and food prices as well as uncertainty, all of which are factors that dampen growth and exacerbate inflationary pressures globally [97,120,121].

Thus, the answer to this problem can be seaweeds because they are aquatic organisms, which permits them to be used as immense and underutilized resources of the seas and oceans, which today provide only up to 2% of human food but span over 70% of the Earth’s surface. Algae farming can help the world meet its goals for decarbonization, zero pollution, circularity, biodiversity preservation and restoration, ecosystem protection and the creation of environmental services. Algae may be used to make plant biostimulants, bio-based chemicals, biofuels and other materials, and they may replace fossil-based goods. Also, as demonstrated in the two World Wars, there is precedence for seaweed as an essential food source for several coastal communities. Thus, nowadays, seaweeds are on the verge of becoming popular due to their suitability as potential feedstock as well as dietary supplements. Because seaweeds are fast-growing, yield high biomass, are elevated and free-of-charge productivity compared to other conventional biomass feedstock, such as maize or soybean, they can meet the growing demand for renewable and sustainable energy sources without compromising food and land resources [121,122,123].

The potential of macroalgae as a renewable resource is currently being studied in Europe. The supply of raw materials and energy is important, and there is a concern regarding the availability of products such as fertilizers, food and energy due to the military aggression between Russia and Ukraine. On the other hand, world population growth also influences the availability of resources and environmental change [124].

The EU (European Union) aims to solve the problems and respond to the growing interest in algae in different industries and economies. Algae are seen as a renewable, competitive and economically, socially and environmentally secure resource for the EU. In addition, macroalgae are seen as an important factor for ecosystems and coastal communities in the regeneration of oceans and waters and are important for current environmental and climate challenges. The macroalgae industry is growing, with developments in knowledge, projects and research. The EU argues that it is necessary to increase the cultivation and production of algae and to develop markets in different industries. In this way, it will be possible to ensure a continuous supply of biomass as well as the smooth running of all the stages and projects involved with algae [124].

## 6. Conclusions

The exploration of seaweed superfoods as bioactive nutraceuticals presents a promising and multifaceted approach to addressing the nutritional challenges and health needs of individuals with celiac disease, diabetes and hyperglycemia. Through the utilization of macroalgae, such as *Ascophyllum nodosum*, *Fucus vesiculosus* and various red algae species like *Chondrus crispus*, *Porphyra*, *Palmaria palmata*, *Gelidium* and *Gracilaria*, innovative food products can be developed with enhanced nutritional content and functional properties. These seaweed-based formulations have shown potential in compensating for the loss of gluten’s characteristics in gluten-free products, controlling glycemic index levels, managing oxidative stress and regulating cholesterol, thus offering valuable support to individuals with celiac disease, diabetes and hyperglycemia. As research in this field progresses, it is envisaged that seaweed superfoods will continue to play a vital role in fostering improved dietary interventions and contributing to the overall well-being of those affected by these conditions.

## Figures and Tables

**Figure 1 marinedrugs-21-00578-f001:**
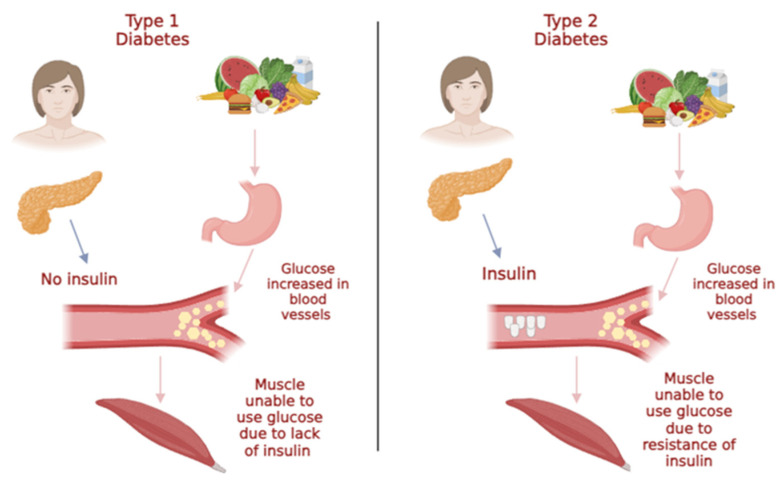
Types of diabetes mellitus and their syndrome.

**Figure 2 marinedrugs-21-00578-f002:**
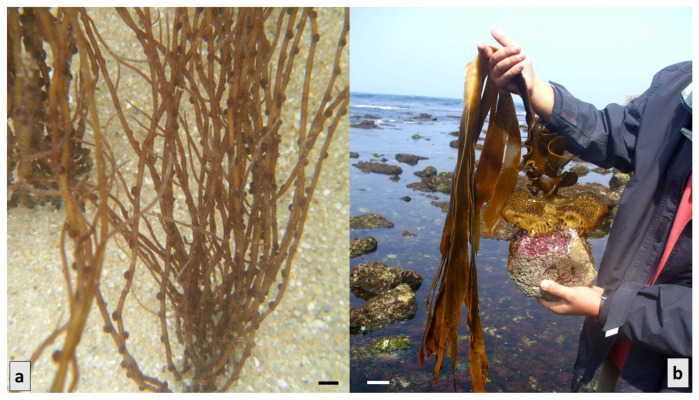
Marine macroalgae: (**a**)*— Gracilaria* sp. (Rhodophyta); (**b**)*—Saccorhiza polyschides* (Phaeophyceae). Scale bar = 1 cm.

**Figure 3 marinedrugs-21-00578-f003:**
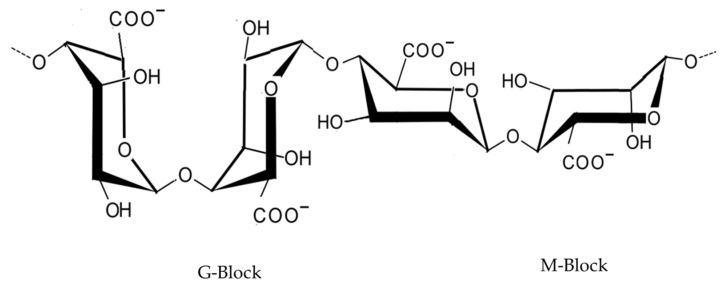
Alginate: alternating α-1,4-l-guluronic acid (G-Block) and β-1,4-d-mannuronic acid (M-Block).

**Figure 4 marinedrugs-21-00578-f004:**
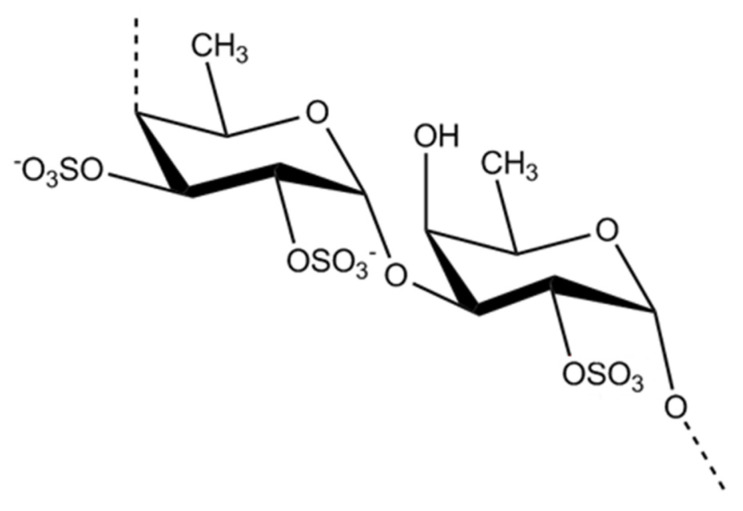
Fucoidan: alternating 1,3- and 1,4-linked α-l-fucopyranose and α-1,3-l-fucopyranose.

**Figure 5 marinedrugs-21-00578-f005:**
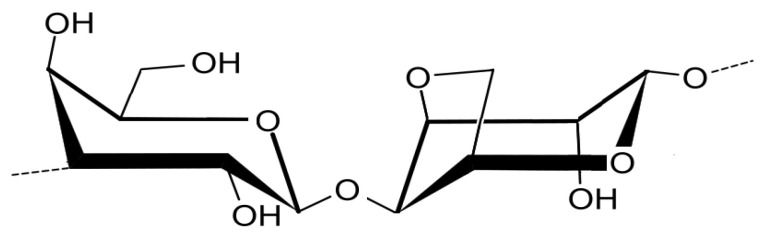
Chemical structure of agar.

**Figure 6 marinedrugs-21-00578-f006:**
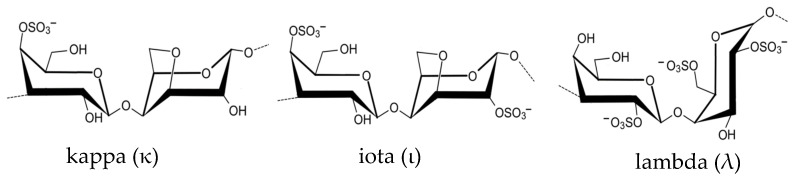
Chemical structure of the three types of carrageenan: kappa, iota and lambda.

**Figure 7 marinedrugs-21-00578-f007:**
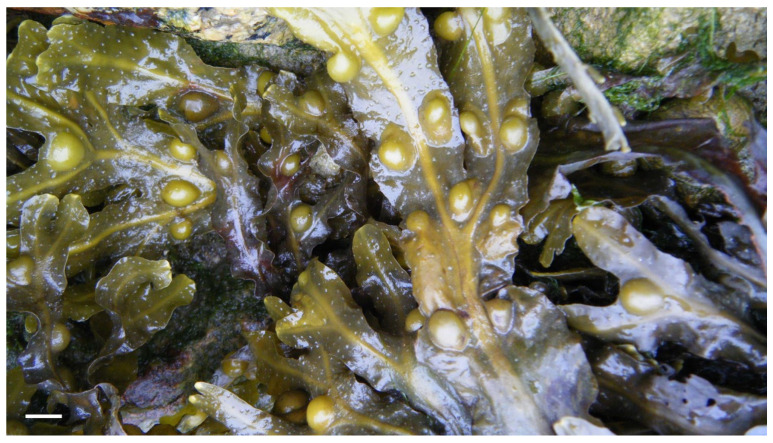
*Fucus vesiculosus*. Scale bar = 1 cm.

**Figure 8 marinedrugs-21-00578-f008:**
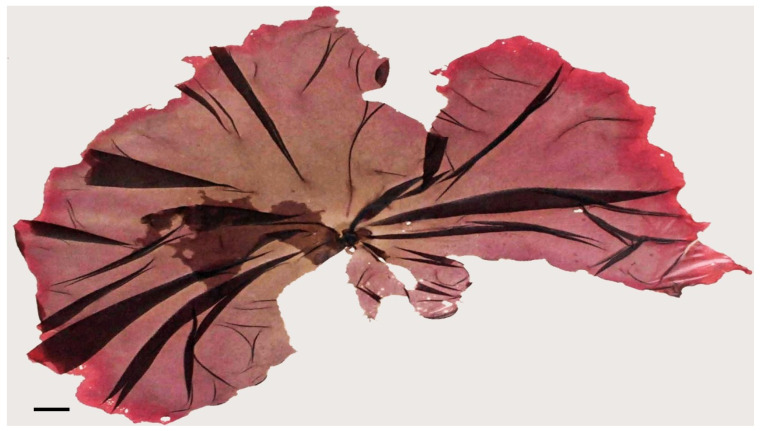
*Porphyra umbilicalis*. Scale bar = 1 cm.

**Figure 9 marinedrugs-21-00578-f009:**
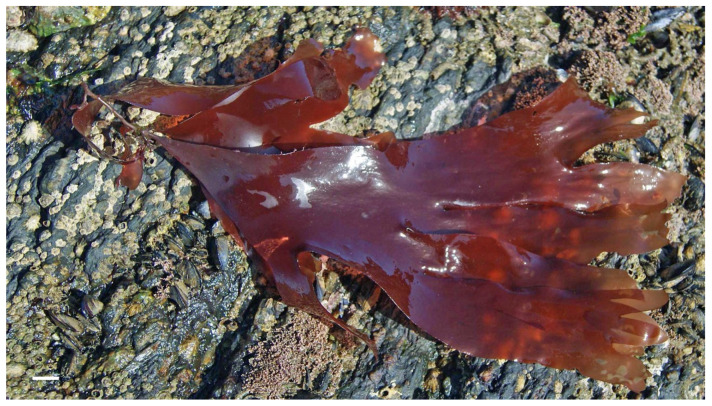
*Palmaria palmata*. Scale bar = 1 cm.

**Figure 10 marinedrugs-21-00578-f010:**
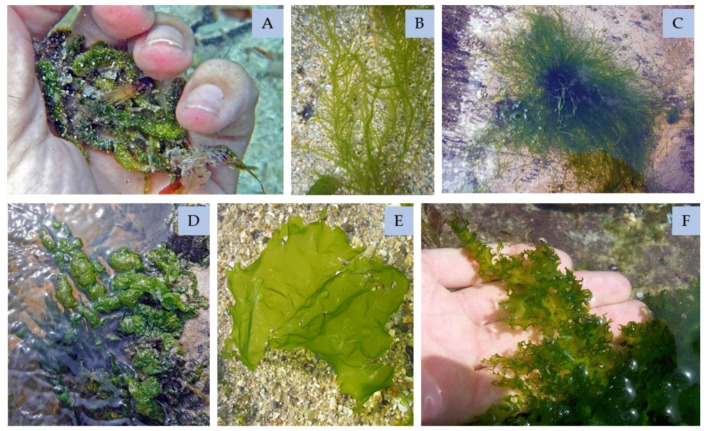
Examples of green algae (Chlorophyta): (**A**)—*Dasycladus vermicularis*; (**B**)—*Ulva clathrata*; (**C**)—*Ulva compressa*; (**D**)—*Ulva intestinal*; (**E**)—*Ulva lactuca*; (**F**)—*Ulva linza* [69].

**Figure 11 marinedrugs-21-00578-f011:**
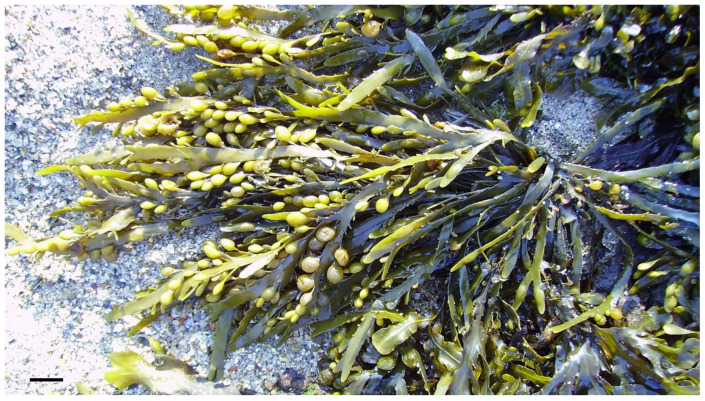
*Ascophyllum nodosum.* Scale bar = 1 cm.

## Data Availability

Not applicable.

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
