# Peer review of "Algae Food Products as a Healthcare Solution"

_marinedrugs, 2023, doi:10.3390/md21110578_

Round 1

Reviewer 1 Report

Comments and Suggestions for Authors

In this review, authors analyze how macroalgae can be game changer for diet-related diseases, though not-healthy food ingredients replacement by macroalgae or macroalgae components, which contain bioactive properties. Thus, observing the potential of macroalgae in diseases such as coeliac disease, diabetes and cholesterol complications.

The paper is well written, with updated references but in my opinion there are a few points I would like to highlight.

1-     The title creates expectations not reflected in the paper; a more appropriate title it could have been; “Algae Food Products as a Healthcare Solution”.

2-     Relevant contribution of this review can be found only in section 4 of the text, previous sections 1-3 provide general information about diet-related diseases and molecules from macroalgae that can be found in many general bibliographic sources.

Based on all above comments, in my opinion this review has not a significant contribution and enough quality to be published in Marine Drugs journal with impact factor: 5.4. I propose this manuscript to be rejected.

Comments on the Quality of English Language

Good Quality of English Language

Author Response

Reviewer 1

Comment 1: The title creates expectations not reflected in the paper; a more appropriate title could have been “Algae Food Products as a Healthcare Solution”.

Answer 1: Thank you for your suggestion. Considering the proposed change, the authors have considered and changed it, adopting the suggested title.

Comment 2: Relevant contribution of this review can be found only in section 4 of the text; previous sections 1-3 provide general information about diet-related diseases and molecules from macroalgae that can be found in many general bibliographic sources.

Answer 2: Thank you for your comments. The authors thought it was important to include sections 3.1-3.3. in the text, as they refer to bioactive compounds present in macroalgae, which will be mentioned later in section 4. In this way, it becomes clearer to understand what is discussed in section 4 with the clarifications in section 3.

To complement the information provided throughout the manuscript, some points have been added (3.3-3.6.) that talk about other bioactive compounds, which are equally important, and where some current studies with these same compounds are mentioned.

Topic 4 was divided into subtopics according to the type of macroalgae, which was supplemented with current information on how the respective seaweeds or their components can be used in different food products.

Finally, we thought it best to add another topic (5.), with current and important information on how macroalgae will be a possible keyway of solving the problems addressed in the manuscript.

Reviewer 2 Report

Comments and Suggestions for Authors

This study is about the rise in gluten-related illnesses and diabetes which highlights the need for better treatments and dietary supplements. It is suggested that seaweed offers potential benefits for both conditions due to its nutritional content and demonstrated abilities to control the glycemic index, reduce oxidative stress, and manage cholesterol.

Based on the title and the content of the study, the abstract section does not reflect the work well. The abstract mainly mentions about the diseases and the significance of microalgae/seaweed was not discussed sufficiently. I suggest the authors to revise the abstract.

The objective of this study is not clearly indicated, and introduction part is poorly written. I recommend the authors revise this section, too. The significance of this study should be emphasized, particularly at the end of the introduction section so that the researchers clearly understand the aim of this work.

In addition, section 3.3 should be revised. The authors mentioned about antioxidants however, this is too general. This part should be enlarged or other sections like 3.4,3.5. etc. should be added as phenolic compounds, carotenoids, etc. These are valuable bioactive compounds and should be discussed in more detail.

Author Response

Reviewer 2

Comment 1: Based on the title and the content of the study, the abstract section does not reflect the work well. The abstract mainly mentions about the diseases and the significance of microalgae/seaweed was not discussed sufficiently. I suggest the authors to revise the abstract.

Answer 1: Taking the proposed suggestion into account, the abstract was revised and modified, and greater emphasis was placed on macroalgae, the bioactive compounds they contain, and how seaweed can be a solution to the problems discussed in the manuscript.

Comment 2: The objective of this study is not clearly indicated, and introduction part is poorly written. I recommend the authors revise this section, too. The significance of this study should be emphasized, particularly at the end of the introduction section so that the researchers clearly understand the aim of this work.

Answer 2: Thank you for your comment. The introductory part has been completely revised and the aim of the work has been changed in the final part of the introduction.

Comment 3: In addition, section 3.3 should be revised. The authors mentioned about antioxidants however, this is too general. This part should be enlarged or other sections like 3.4,3.5. etc. should be added as phenolic compounds, carotenoids, etc. These are valuable bioactive compounds and should be discussed in more detail.

Answer 3: We appreciate your comments, and the authors agreed. The suggested compounds, which are equally important and referred to throughout the manuscript, have been added, so it makes sense for them to be included in the text, also for a better understanding by researchers of the various subjects covered.

Editor: As suggested, the authors prepared a Graphical Abstract

Round 2

Reviewer 1 Report

Comments and Suggestions for Authors

I would like to know why you have sent me the article that I previously reviewed. My recommendation was to reject it, and I do not know why I should review it again. If reviewer 1 recommends to be accepted, I have no objection, but please respect my work and evaluation, as I have sufficient expertise as reviewer.

Comments on the Quality of English Language

Minor editing of English language required

Author Response

Comment 1: To present the term Nutraceutic since the manuscript combine food and health care topics. In the line 399 the term novel food is introduced. I suggest developing this aspect with more details, advances and problems related to this, concerning the use of seaweed in Europe as novel food, a few sentences on legal limitations. I am not suggesting large text but at least several paragraphs since there are already previous reports on this topic.

Answer 1: Thank you for your suggestion. A definition of nutraceutical has been added, which fits the manuscript and defines the term well. A point (5.2.) has been added that addresses the problems, advantages, and limitations of using seaweed as novel foods.

Comment 2: In the point 5.4 Future Road discuss on problems concerning the use of algae as food, the strategy to expand the seaweeds but also microalgae as food in non-Asiatic countries can be more extended treated including comment of the European strategy. Communication (2022):592 Towards a strong and sustainable EU algae sector Porphyra/Pyropia species can be expressed as Porphyra sensu lato.

Answer 2: Thank you for your comments. Point 5.5 (Future Road) has been completely revised, taking into account the suggestions made. Two paragraphs have been added on how the EU wants to use macroalgae for current global problems, and the benefits of improving the entire macroalgae industry.

Comment 3: The point 3.7 Natural antioxidants chapter is too general and there is a certain repetition since polyphenols are cited in 3.3, ) however in 3.7 other important antioxidant substances as carotenoids are not included but it is cited 3.5. Mycosporyne like aminoacids other important antioxidant substances in red macroalgae are not cited. As suggest as follows: To delete the general chapter on natural antioxidants (3.7) and integrate the antioxidant function separately in each point on specific bioactive compounds i.e 3.3 Polyphenols, 3.4 Pigments, 3.5 Carotenoids 3.6 Fatty acids and to open new point 3.7 Mycosporyne like amino acids.

Answer 3: Considering the proposed suggestion, a point (3.7.) Mycosporyne-like amino acids have been added. Thank you for your suggestion.